# Insight into the Wnt Pathway in Sporadic Small Bowel Adenocarcinoma

**DOI:** 10.3390/cancers17182965

**Published:** 2025-09-10

**Authors:** Takayoshi Nishimoto, Atsushi Tatsuguchi, Takeshi Yamada, Sho Kuriyama, Aitoshi Hoshimoto, Jun Omori, Naohiko Akimoto, Katya Gudis, Keigo Mitsui, Shu Tanaka, Shunji Fujimori, Tsutomu Hatori, Akira Shimizu, Masanori Atsukawa

**Affiliations:** 1Department of Gastroenterology, Nippon Medical School, Tokyo 113-8603, Japan; clubman629@nms.ac.jp (T.N.); a-hoshimoto@nms.ac.jp (A.H.); 67trocadero@nms.ac.jp (J.O.); s03-004an@nms.ac.jp (N.A.); kgudis@gmail.com (K.G.); mitsui@nms.ac.jp (K.M.); tanashu@nms.ac.jp (S.T.); s-fujimori@nms.ac.jp (S.F.); gachi@nms.ac.jp (M.A.); 2Department of Analytic Human Pathology, Nippon Medical School, Tokyo 113-8602, Japan; ashimizu@nms.ac.jp; 3Department of Gastrointestinal and Hepato-Biliary-Pancreatic Surgery, Nippon Medical School, Tokyo 113-8603, Japan; y-tak@nms.ac.jp (T.Y.); mcfxt875@nms.ac.jp (S.K.); 4Department of Pathology, Nippon Medical School, Chiba Hokusoh Hospital, 1715 Kamagari, Inzai 270-1694, Chiba, Japan; hatorittm@nms.ac.jp

**Keywords:** *APC* mutation, β-catenin, genomic profile, small bowel adenocarcinoma, Wnt pathway, Wnt5a

## Abstract

We investigated the involvement of the Wnt signaling pathway in small bowel adenocarcinoma (SBA) by examining genomic alterations and the expression of downstream target gene products or associated proteins. Mutations in *APC* and *CTNNB1*, key components of the Wnt pathway, were identified in 23% of cases. Aberrant expression of β-catenin, a central molecule in the canonical Wnt pathway, was observed in 37% of cases, suggesting pathway activation. Cyclin D1 and c-Myc, known Wnt pathway target proteins, were positively expressed in 60% and 41% of cases, respectively. Aberrant expression of β-catenin and/or Wnt5a, a ligand of the noncanonical Wnt pathway, was detected in 60% of cases and showed a correlation with both cyclin D1 and c-Myc expression. These findings indicate that both canonical and noncanonical Wnt pathway-related proteins contribute to carcinogenesis and tumor progression in SBA.

## 1. Introduction

Small bowel cancers are rare, representing less than 5% of gastrointestinal cancers, though their incidence is rising [1,2]. Small bowel adenocarcinoma (SBA) comprises roughly one-third of these cases. Due to its rarity, detailed molecular and clinicopathological data remain limited [1,3,4]. Genomic profiling has revealed that SBA has distinct molecular features compared to colorectal and gastric cancers [5]. Additionally, differences in its molecular and clinicopathological features have been reported across small bowel subsites [3,6,7,8]. For example, duodenal adenocarcinomas have higher rates of *CDKN2A* and *ERBB2* alterations, but lower incidences of *BRAF*, *PTEN*, and *PIK3R1* mutations, as well as a lower overall tumor mutational burden, compared to jejunal and ileal adenocarcinomas [5,9].

It remains uncertain whether carcinomas originating throughout the duodenum, jejunum, and ileum share common molecular characteristics and carcinogenetic pathways. Preinvasive epithelial lesions in the small bowel are less frequently observed than in the colon, and are predominantly located in the duodenum. Sporadic non-ampullary duodenal adenomas, excluding familial adenomatous polyposis and other predisposing conditions, account for 40% of such lesions [10,11]. Okada et al. reported a 4.7% progression rate from sporadic adenoma to non-invasive carcinoma [12]. While activation of the Wnt signaling pathway is characteristic of the adenoma–carcinoma sequence, the degree and nature of its involvement in SBA remains uncertain and controversial.

The Wnt signaling pathway can be divided into canonical and noncanonical pathways [13]. The canonical Wnt pathway involves cytosolic β-catenin accumulation, nuclear translocation, and interaction with TCF/LEF transcription factors to induce the expression of various oncoproteins, such as cyclin D1 and c-Myc [14,15]. In contrast to canonical Wnt/*β*-catenin signaling, the noncanonical (β-catenin-independent) Wnt pathway, including Wnt/Ca^2+^ and planar cell polarity subtypes, is less studied in carcinogenesis. Wnt5a, a key noncanonical Wnt ligand, activates downstream signaling through Frizzled receptors, influencing cell migration and adhesion [16].

To date, the functional status and clinicopathological relevance of canonical and noncanonical Wnt pathways in SBA have not been thoroughly elucidated. In this study, we investigated the correlation between Wnt pathway-related genomic alterations and the expression of downstream target gene products or associated proteins to better understand their roles in SBA carcinogenesis.

## 2. Materials and Methods

### 2.1. Patients and Tissue Samples

The flowchart of the study is presented in Figure 1. We analyzed 75 tissue samples of duodenal, jejunal, and ileal adenocarcinomas obtained from the archives of the Department of Pathology at Nippon Medical School Hospital and the Department of Pathology at Nippon Medical School Chiba Hokusoh Hospital between January 2006 and December 2022. These samples were used for immunohistochemical analysis of β-catenin, cyclin D1, c-Myc, E-cadherin, and Wnt5a expression. To focus on sporadic small bowel adenocarcinoma (SBA), patients with predisposing conditions—such as Lynch syndrome, familial adenomatous polyposis, celiac disease, Crohn’s disease, or Peutz–Jeghers syndrome—were excluded. Cases of ampullary adenocarcinomas and suspected metastatic cancers were also excluded. The cancer-specific survival (CSS) was defined as the interval from the date of initial surgery to death caused by SBA, excluding other causes. All the patients provided informed consent, and the study was approved by the Ethics Committee of Nippon Medical School. Tumor staging was based on the TNM classification by the International Union Against Cancer.

### 2.2. Immunohistochemical Analysis

Tissue specimens were fixed in 10% formalin (MUTO PURE CHEMICALS, Tokyo, Japan), embedded in paraffin wax, and deparaffinized. Endogenous peroxidase activity was blocked by immersion in 0.5% H_2_O_2_–methanol (FUJIFILM Wako, Osaka, Japan) for 10 min. Antigen retrieval was performed using microwave heating in either 0.01 mol/L citrate phosphate buffer (Vector Laboratories, CA, USA) (pH 6.0) or EDTA (Vector) (pH 9.0), followed by incubation with 10% normal horse or goat serum (Vector) to block nonspecific binding. Slides were incubated for 18 h at 4 °C with primary antibodies (Dako, Agilent, CA, USA) (listed in Appendix A). Secondary detection was performed using biotinylated anti-mouse IgG or anti-rabbit IgG antibodies (Vector) (1:200, Vector) at 25 °C for 30 min, followed by avidin–biotin–peroxidase complex (Vector) for another 30 min at same temperature. Visualization was achieved using 3,3′-diaminobenzidine tetrahydrochloride solution containing 0.03% H_2_O_2_ (NICHIREI BIOSCIENCES, Tokyo, Japan).

### 2.3. Evaluation of Immunohistochemical Staining

All the slides were independently evaluated by two observers (A.H. and A.S.) who were blinded to the clinical data. Discrepancies were resolved using a multi-headed microscope. The staining intensity was scored based on the proportion of positively stained epithelial cells, using the following 5-point scale: 0, <10%; 1, 10–25%; 2, 26–50%; 3, 51–75%; and 4, >75%. For β-catenin, cyclin D1, and c-Myc, the cases with nuclear staining in ≥10% of cancer cells (score 1–4) were considered positive [17]. For E-cadherin and β-catenin, the expression was considered preserved if the membranous staining was equivalent to that of a normal small intestinal epithelium in >75% of cancer cells (score 4), and reduced if less than this threshold (score 0–3). The mucin immunophenotypes were categorized as gastric, intestinal, gastrointestinal, or null types based on the immunostaining for MUC2, MUC5AC, MUC6, and CD10. MUC 2 and CD10 were considered intestinal markers, while MUC5AC and MUC6 were considered gastric markers. The tumors expressing both gastric and intestinal markers were classified as gastrointestinal type, and those expressing neither were classified as null type.

The DNA mismatch repair (MMR) status was assessed via immunostaining for MLH1, MSH2, MLH6, and PMS2. The tumors lacking expression of any one MMR protein were categorized as MMR deficient (dMMR); all others were considered MMR proficient (pMMR).

### 2.4. Next-Generation Sequencing (NGS)

NGS was performed on 48 patients with sufficient DNA extracted from paraffin-embedded tissue blocks. DNA extraction, amplification, and sequencing were conducted as previously described [18,19,20]. Briefly, approximately 10 ng of DNA per sample was amplified using multiplex PCR with Ion AmpliSeq Cancer Hotspot Panel v2 (Thermo Fisher Scientific, Waltham, MA, USA), which targets 207 amplicons covering hotspot regions for 50 commonly mutated oncogenes and tumor suppressor genes as cataloged in COSMIC. Sequencing was performed on Ion Torrent PGM™ system (Thermo Fisher Scientific), and data were analyzed using Torrent Suite™ Software v5.2.2. Variant calling was carried out using CHP2 Panel Somatic PGM under low stringency settings. Only variants classified as pathogenic or likely pathogenic according to ClinVar were included in final analysis.

### 2.5. Statistical Analysis

The associations between the immunohistochemical findings and clinicopathological features were evaluated using a chi-square test or Fisher’s exact test, as appropriate. When multiple comparisons were conducted, a Bonferroni correction was applied. The correlations between protein expressions were assessed using the same statistical tests. The impact of clinicopathological variables on the cancer-specific survival (CSS) was assessed using the Cox proportional hazards model. The variables with *p* < 0.05 in the univariate analysis were included in the multivariate model. A *p*-value < 0.05 was considered statistically significant.

## 3. Results

### 3.1. Demographic Data of Patients

The study included 75 patients, comprising 53 men and 22 women, with ages ranging from 32 to 84 years (mean: 65 years; median: 68 years). The tumors were located in the duodenum in 40 cases, the jejunum in 30 cases, and the ileum in 5 cases. The mucin immunophenotypes were classified as gastric (*n* = 14), gastrointestinal (*n* = 30), intestinal (*n* = 29), and null type (*n* = 2). At the time of analysis, 22 patients had died. The overall 5-year survival rate was 71%. The median follow-up duration was 52 months (mean: 50 months; range, 5–115 months).

### 3.2. Localization of β-Catenin, Cyclin D1, c-Myc, E-Cadherin, and Wnt5a in SBA

The immunostaining patterns for β-catenin, cyclin D1, c-Myc, E-cadherin, and Wnt5a are illustrated in Figure 2. The quantitative immunoreactivity data, based on the percentage of unequivocally positive epithelial cells, are presented in Appendix A.

Reduced membranous expression of β-catenin, often accompanied by cytoplasmic localization in tumor cells, was observed in 28 cases (37%). Among these, nuclear expression of β-catenin was identified at the invasive tumor front in nine cases (13%). Cyclin D1 expression was exclusively nuclear in 45 cases (60%). Similarly, c-Myc expression was observed exclusively in the nuclei of tumor cells in 31 cases (41%). A reduced membranous expression of E-cadherin in cancer cells occurred in 33 cases (44%). Wnt5a was expressed in the membranes and cytoplasm of tumor cells in 35 cases (47%), but was nearly undetectable in the non-neoplastic epithelium. Sporadic Wnt5a positivity was also noted in some stromal cells, including fibroblasts.

### 3.3. Association Between Marker Expression and Clinicopathological Factors

The relationships between the immunohistochemical findings and clinicopathological features are summarized in Table 1.

β-Catenin: Its reduced membranous expression was significantly associated with lymph node metastasis, distant metastasis, and an advanced TNM stage. However, the nuclear expression of β-catenin showed no significant correlation with the clinicopathological parameters.

Cyclin D1: Its expression was more frequent in cases with lymph node metastasis and less commonly observed in gastric-type SBA.

E-cadherin: A reduced membranous expression of E-cadherin was more common in poorly differentiated adenocarcinoma and was significantly correlated with the histological subtype, depth of invasion (pT factor), lymph node metastasis, distant metastasis, and an advanced TNM stage.

Wnt5a: The expression was significantly correlated with the pT factor, lymph node metastasis, distant metastasis, and a higher TNM stage.

### 3.4. Interrelationship Between Marker Expressions

The interrelationships between β-catenin, cyclin D1, c-Myc, E-cadherin, and Wnt5a expression are presented in Table 2.

A significant positive correlation was found between reduced membranous β-catenin expression and both cyclin D1 and c-Myc. The cyclin D1 and c-Myc expressions were also significantly positively correlated with each other. A reduced membranous E-cadherin expression was associated with a reduced membranous β-catenin expression, cyclin D1 positivity, and Wnt5a expression. In total, 60% of cases exhibited reduced membranous β-catenin expression and/or Wnt5a expression, which was significantly associated with increased expression of cyclin D1 and c-Myc.

### 3.5. Association Between Gene Mutations and Marker Expression

The NGS results are provided in Appendix A. The genomic alterations were identified in 40 of the 48 cases (83%). The most frequently mutated genes were *TP53*, 47.9%; *KRAS*, 31.3%; *APC*, 14.6%; and *CTNNB1*, 8.3%. The detailed correlations between the mutations and clinicopathological parameters are outlined in Appendix A. *TP53* mutations were more prevalent in the tumors located in the jejunum. *APC* and *CTNNB1* mutations were exclusively identified in intestinal- and gastrointestinal-type SBA, with none detected in gastric-type SBA. No single case harbored mutations in both *APC* and *CTNNB1*. All four cases with *CTNNB1* mutations exhibited reduced membranous β-catenin expression. In contrast, only one of the seven *APC* mutation cases showed a similar β-catenin reduction (Figure 3). Among the 20 cases with reduced β-catenin expression, 13 (patients 36–48) had neither *APC* or *CTNNB1* mutations. Of the seven *APC* mutant cases, cyclin D1 was positive in three cases (patients 29–31) and c-Myc was positive in two cases (patients 30 and 31). The remaining three cases (patients 25–28) were negative for both cyclinD1 and c-Myc, but showed *TP53* mutations and a mismatch repair (MMR) deficiency. Of the four *CTNNB1* mutant cases, cyclin D1 was positive in three cases (patients 33–35) and c-Myc was positive in two cases (patients 34 and 35). All four tumors had a proficient MMR status, with two lacking *TP53* and *KRAS* mutations.

### 3.6. Comparative Survival Analysis

The pT factor; lymph node metastasis; and expression of β-catenin, E-cadherin, and Wnt5a were significantly associated with the CSS (Table 3). No single gene mutation showed a statistically significant association with the CSS. In the multivariate analysis incorporating all the variables into the Cox proportional hazards, the lymph node status emerged as the only independent prognostic factor. However, in an alternative model that included the pT factor, lymph node status, and reduced membranous expression of E-cadherin and β-catenin, both the lymph node status and reduced membranous expression of E-cadherin and β-catenin retained independent prognostic significance (Table 3).

## 4. Discussion

To elucidate the carcinogenetic pathway in SBA, we performed a comprehensive analysis of gene mutations and evaluated the localization of proteins that were associated with these gene mutations, along with their correlation with clinicopathological features. Given the relatively low incidence of *APC* mutations (7.1–37%) in sporadic SBA, in contrast to their high frequency in adenomas, the adenoma–carcinoma sequence may not constitute the predominant pathway in SBA carcinogenesis [21,22,23,24]. The gene mutations linked to the Wnt pathway in SBA included *APC* and *CTNNB1*, with one or the other present in 23% of cases. Notably, only one of seven SBA cases with *APC* mutations exhibited reduced membranous β-catenin expression, suggesting limited activation of the canonical Wnt pathway even in the presence of *APC* mutations. Furthermore, six of these seven *APC*-mutant cases also harbored *TP53* mutations, and three showed MMR deficiencies, implying that *APC* mutations may not independently drive SBA carcinogenesis. Conversely, β-catenin staining abnormalities—an indicator of canonical Wnt pathway activation—were observed in 37% of SBA cases. Previous studies have reported abnormal β-catenin staining in 7–93% of SBA cases, and nuclear localization in 7–48%, which aligns with our findings [21,22,25]. These studies, like ours, observed a lack of correlation between abnormal β-catenin expression and *APC* mutations. The current study shows higher rates of β-catenin expression abnormalities than was found for the combined mutation rates of *APC* and *CTNNB1*. Moreover, we found higher rates of cyclin D1 and c-Myc expression than *APC*/*CTNNB1* mutations, indicating downstream activation of Wnt target genes despite the absence of these mutations. To address this discrepancy, we categorized the SBAs into gastric and intestinal types based on the histologic phenotype, since prior research suggests that these subtypes may arise through distinct molecular pathways [11]. Ota et al. reported that the canonical Wnt pathway is less involved in gastric-type non-ampullary duodenal tumors, which exhibit lower rates of nuclear β-catenin expression and *APC* mutations [23]. Our results support this finding: *APC* and *CTNNB1* mutations were exclusive to intestinal-type SBA, with a combined mutation rate of 39.1% (9/23), closely matching the 44.8% (13/29) abnormal β-catenin expression rate in this subtype (Figure 4). These findings suggest that the canonical Wnt pathway contributes predominantly to intestinal-type SBA, though *APC* mutations alone may be insufficient to drive the progression from adenoma to adenocarcinoma.

No prior studies have examined the relationship between the Wnt pathway target proteins and β-catenin abnormalities in SBA. We found that c-Myc was expressed in 41% of cases and was significantly correlated with abnormal β-catenin expression, indicating its upregulation via the canonical Wnt pathway. Cyclin D1 was expressed in 60% of cases, paralleling the findings of a previous study [26]. Although cyclin D1 expression showed a strong correlation with abnormal β-catenin expression, nearly half of the cyclin D1-positive cases lacked β-catenin abnormalities, suggesting that other regulatory pathways may also influence cyclin D1 expression.

Given the gap between the relatively low frequency of *APC*/*CTNNB1* mutations and the high frequency of Wnt pathway activation, we hypothesized that the noncanonical Wnt pathway may also play a role in SBA carcinogenesis. We therefore evaluated the localization of Wnt5a, a ligand of the noncanonical Wnt pathway, by immunohistochemistry [27]. In colorectal cancer, Wnt5a has been reported to act as either an oncogene or a tumor suppressor, depending on the downstream pathway activation [28]. Wnt5a can modulate both canonical and noncanonical Wnt signaling [29,30], and its prognostic value appears to be context-dependent [28]. For example, Wnt5a has been shown to decrease Lgr5/RSPO3 expression and β-catenin activity in vitro, and its expression has been associated with improved survival in colon cancer [31]. In contrast, other studies have reported that Wnt5a promotes epithelial–mesenchymal transition and contributes to colorectal cancer progression and metastasis [32]. Similarly, Wnt5a gene expression has been associated with better overall survival in lung squamous cell carcinoma, but with a poor prognosis in gastric adenocarcinoma [33]. In our cohort, Wnt5a was expressed in 46.7% of SBA cases, primarily in cancer cells. To our knowledge, this is the first study to investigate Wnt5a localization in SBA and its association with the clinicopathological parameters. We found that Wnt5a expression was significantly correlated with the pT factor, metastasis, and a poor prognosis, independent of mucin phenotype or tumor location. Moreover, Wnt5a expression was correlated with cyclin D1 and E-cadherin localization, but not with β-catenin or c-Myc. The observed correlation between Wnt5a and E-cadherin downregulation supports this mechanism and highlights the potential involvement of β-catenin-independent Wnt pathway in SBA progression (Figure 4).

Two earlier studies reported reduced membranous E-cadherin expression in 38% and 41.8% of SBA cases, respectively, slightly lower than our findings [21,25]. These variations may reflect differences in the proportion of advanced-stage tumors. In our cohort, E-cadherin downregulation was associated with β-catenin abnormalities and was correlated with a poor prognosis. These findings are consistent with those of Lee et al., who also reported that E-cadherin and β-catenin expression are significant worse prognostic factors in SBA [25]. Notably, E-cadherin loss was not correlated with any genomic alterations, but was significantly associated with protein-level β-catenin abnormalities.

Although prior studies have suggested that the small bowel subsite (duodenum, jejunum, or ileum) may influence the molecular features [3,6,7,8,34,35], we found no significant association between the tumor location and genomic alterations, except for the *TP53* mutations, which were more common in jejunal tumors. Instead, our results indicate that the histologic phenotype, rather than the anatomical location, better reflects the molecular and clinicopathological characteristics of SBA.

This study has several limitations. First, the small sample size limited the number of cases available for next-generation sequencing. In addition, there was a bias in the tumor location. Half of the cases in this study were located in the duodenum. As the duodenum is the most common site of small bowel adenocarcinoma, this distribution was expected. However, the relatively high proportion of stage I cases in the duodenum may represent a potential source of bias. Nevertheless, because no significant differences in the Wnt pathway molecules were observed by subsite, this is unlikely to affect the generalizability of our findings. Second, the MMR status was assessed solely via immunohistochemistry, without MMR testing. Third, the Cancer Hotspot Panel v2 used included only a subset of genes, potentially missing other relevant Wnt pathway alterations.

## 5. Conclusions

In conclusion, canonical Wnt signaling appears to be a key driver of carcinogenesis in intestinal-type SBA, whereas noncanonical Wnt signaling may contribute to tumor progression across SBA subtypes. We propose that activation of the Wnt pathway has potential value for predicting treatment response and prognosis, as a reduced expression of E-cadherin and β-catenin is associated with poorer outcomes, and the Wnt5a may represent a therapeutic target. Future studies should examine whether targeting the Wnt5a or restoring E-cadherin function could provide clinical benefits.

## Figures and Tables

**Figure 1 cancers-17-02965-f001:**
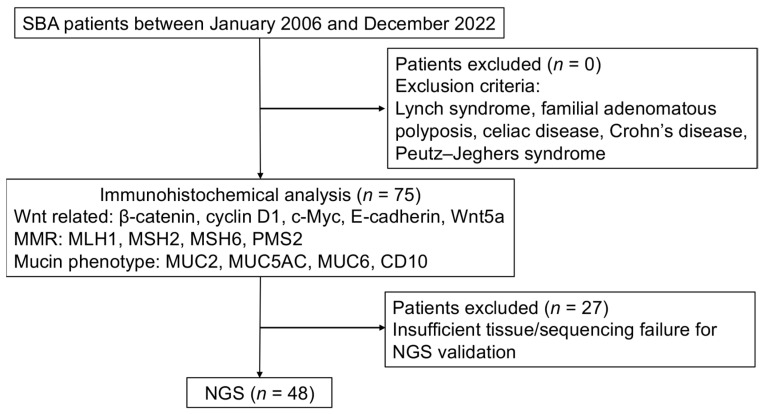
Flowchart of this study. A total of 75 SBA patients were enrolled and underwent immunohistochemical analysis using FFPE blocks. NGS was performed on 48 patients using the same FFPE blocks. Abbreviations: FFPE, formalin-fixed paraffin-embedded; MMR, DNA mismatch repair; NGS, next-generation sequencing; SBA, small bowel adenocarcinoma.

**Figure 2 cancers-17-02965-f002:**
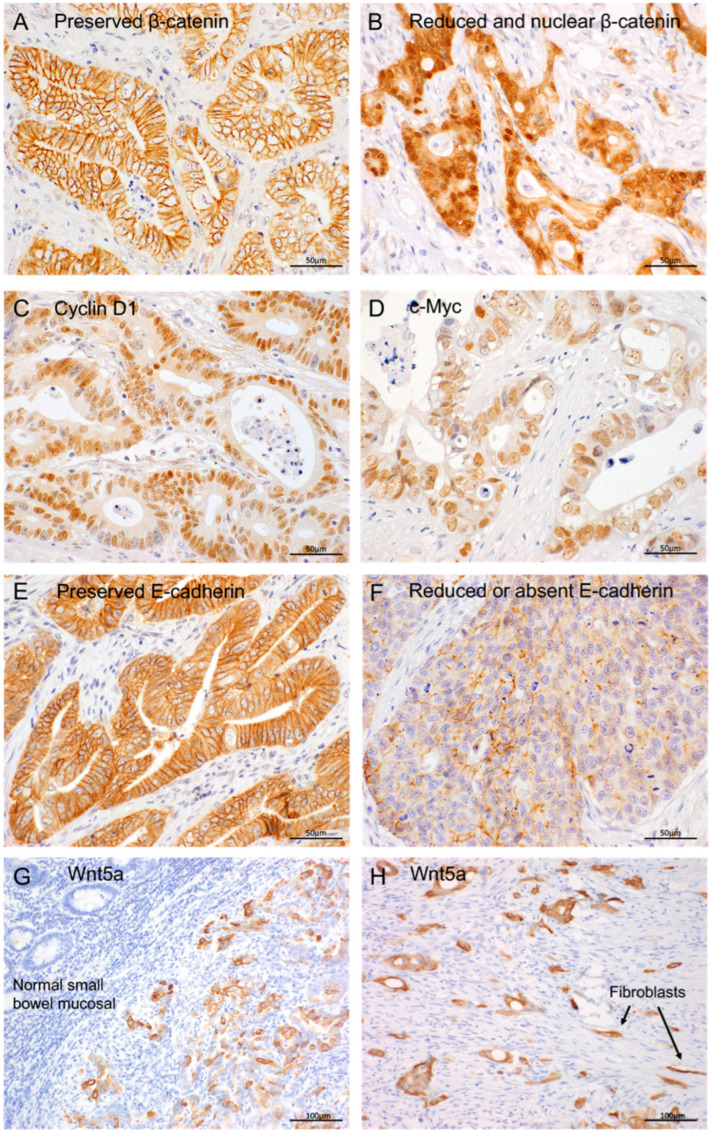
Immunohistochemical localization of β-catenin (**A**,**B**), cyclin D1 (**C**), c-Myc (**D**), E-cadherin (**E**,**F**), and Wnt5a (**G**,**H**) in small bowel adenocarcinoma. (**A**) β-catenin is localized to the cytoplasmic membrane of adenocarcinoma cells. (**B**) β-catenin is mislocalized to the cytoplasm and nuclei of adenocarcinoma cells, indicating activation of the Wnt pathway. (**C**) Cyclin D1 is detected in the nuclei of adenocarcinoma cells. (**D**) C-Myc is also observed in the nuclei of adenocarcinoma cells. (**E**) E-cadherin is localized to the cytoplasmic membrane of adenocarcinoma cells. (**F**) Reduced or absent membranous E-cadherin staining in adenocarcinoma cells. (**G**) Wnt5a is detected in the cytoplasmic membrane and cytoplasm of adenocarcinoma cells, but not in normal small bowel mucosal epithelial cells. (**H**) A few fibroblasts in cancer stroma are also positive for Wnt5a. Original magnification: 40×, (**A**–**F**); 20×, (**G**,**H**).

**Figure 3 cancers-17-02965-f003:**
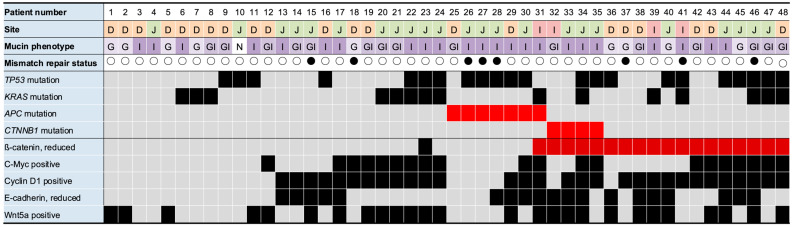
Summary of gene mutations and clinicopathological features in 48 patients with small bowel adenocarcinoma. Cases showing reduced membranous β-catenin expression—suggesting activation of Wnt pathway—are indicated in dark red. *APC/CTNNB1* mutations are indicated in light red. DNA mismatch repair (MMR)-deficient cases are marked with black circles, and MMR-proficient cases with white circles. D, duodenum; J, jejunum; I, ileum. Mucin phenotype: G = gastric type; GI = gastrointestinal type; I = intestinal type; N = null type.

**Figure 4 cancers-17-02965-f004:**
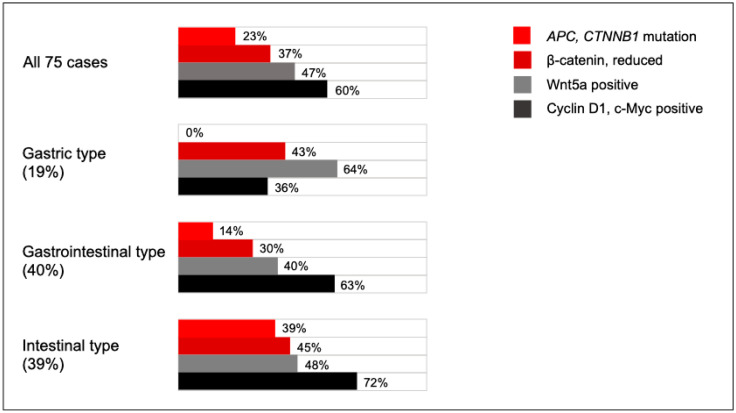
Graphical abstract summarizing the results and conclusions of this study.

**Table 1 cancers-17-02965-t001:** Relationship between cyclin D1, c-Myc, β-catenin, E-cadherin, Wnt5a, and clinicopathologic factors (*n* = 75).

		Cyclin D1		C-Myc		Reduced β-Catenin		Reduced E-Cadherin		Wnt5a	
	**No.**	**No. (%)**	** *p* **	**No. (%)**	** *p* **	**No. (%)**	** *p* **	**No. (%)**	** *p* **	**No. (%)**	** *p* **
Age											
<69	35	22 (63)	NS	19 (58)	0.038	15 (43)	NS	14 (40)	NS	18 (51)	NS
≥69	40	23 (58)		12 (30)		13 (33)		19 (48)		17 (43)	
Sex											
Female	22	12 (55)	NS	9 (41)	NS	8 (36)	NS	11 (50)	NS	11 (50)	NS
Male	53	33 (62)		22 (42)		20 (38)		22 (42)		24 (45)	
Site											
Duodenum	40	18 (45)	0.018	13 (33)	NS	11 (28)	0.006	11 (28)	0.009	15 (38)	NS
Jejunum	30	23 (77)		17 (57)		12 (40)		19 (63)		18 (60)	
Ileum	5	4 (80)		1 (20)		5 (100)		3 (60)		2 (40)	
Histologic Type											
WD, MD	60	36 (60)	NS	24 (40)	NS	20 (33)	NS	22 (37)	0.029	27 (45)	NS
PD	10	7 (70)		6 (60)		5 (50)		8 (80)		4 (40)	
Mucinous	5	2 (50)		1 (20)		3 (60)		3 (60)		4 (80)	
Depth of invasion (pT Factor)											
pT1–3	62	36 (58)	NS	24 (39)	NS	20 (32)	NS	23 (37)	0.013	24 (39)	0.004
pT4	13	9 (69)		7 (54)		8 (62)		10 (77)		11 (85)	
Lymph Node Metastasis											
N0	47	24 (51)	0.053	17 (36)	NS	10 (21)	<0.001	9 (19)	<0.001	12 (26)	<0.001
Nx	28	21 (75)		14 (50)		18 (64)		24 (86)		23 (82)	
Distant Metastasis											
M0	59	33 (56)	NS	24 (41)	NS	16 (27)	0.001	17 (29)	<0.001	23 (39)	0.013
Mx	16	12 (75)		7 (44)		12 (75)		16 (100)		12 (75)	
TNM Stage											
l	26	13 (50)	NS	7 (27)	NS	6 (23)	0.001	4 (15)	<0.001	3 (12)	<0.001
II	21	11 (52)		10 (48)		4 (19)		5 (24)		9 (43)	
III	12	9 (75)		7 (58)		6 (50)		8 (67)		11 (92)	
IV	16	12 (75)		7 (44)		12 (75)		16 (100)		12 (75)	
MMR Status											
Proficient	65	39 (60)	NS	27 (42)	NS	25 (39)	NS	28 (43)	NS	31 (28)	NS
Deficient	10	6 (60)		4 (40)		3 (33)		5 (50)		4 (40)	
Mucin Phenotype											
Gastric type	14	5 (36)	0.038	4 (29)	NS	6 (43)	NS	7 (50)	NS	9 (64)	NS
Gastrointestinal type	30	19 (63)		14 (47)		9 (30)		8 (27)		12 (40)	
Intestinal type	29	21 (72)		13 (45)		13 (45)		17 (59)		14 (48)	
Null type	2	0		0		0		1 (50)		0	

MD, moderately differentiated tubular adenocarcinoma; mucinous, mucinous adenocarcinoma; MMR, DNA mismatch repair; PD, poorly differentiated tubular adenocarcinoma; WD, well-differentiated tubular adenocarcinoma; NS, not significantly different.

**Table 2 cancers-17-02965-t002:** Mutual relationship between cyclin D1, c-Myc, β-catenin, E-cadherin and Wnt5a expression (*n* = 75).

	Cyclin D1	C-Myc	E-Cadherin	Wnt5a
β-Catenin	0.002	0.022	0.003	NS
Cyclin D1		0.001	0.001	0.021
C-Myc			NS	NS
E-cadherin				0.003

Data is given by *p* value assessed using chi-square test or Fisher’s exact test, as appropriate. NS, not significantly different.

**Table 3 cancers-17-02965-t003:** Univariate and multivariate Cox proportional hazards analysis for cancer-specific survival (Stage I-IV) (*n* = 75).

Variables	Categories	Hazard Ratio (95% CI)	*p* Value
Univariate Analysis			
Histologic Type	PD, mucinous vs. WD, MD	2.248 (0.915–5.523)	0.077
pT Factor	pT4 vs. pT1–3	4.021 (1.676–9.646)	0.002
Lymph Node Metastasis	positive vs. negative	26.87 (6.253–115.47)	<0.001
β-Catenin	reduced vs. preserved	3.720 (1.558–8.886)	0.003
Cyclin D1	positive vs. negative	1.256 (0.527–2.995)	NS
C-Myc	positive vs. negative	1.275 (0.549–2.958)	NS
Wnt5a	positive vs. negative	4.207 (1.640–10.79)	0.003
E-Cadherin	reduced vs. preserved	10.56 (3.122–35.75)	<0.001
β-Catenin/E-Cadherin	both reduced vs. others	8.155 (3.398–19.58)	<0.001
Multivariate Analysis			
pT Factor	pT4 vs. pT1–3	1.258 (0.512–3.096)	NS
Lymph Node Metastasis	positive vs. negative	16.52 (3.494–78.12)	<0.001
β-Catenin/E-cadherin	both reduced vs. others	2.576 (1.020–6.507)	0.045

CI, Confidence Interval; MD, moderately differentiated tubular adenocarcinoma; mucinous, mucinous adenocarcinoma; PD, poorly differentiated tubular adenocarcinoma; pT factor, Depth of invasion; WD, well-differentiated tubular adenocarcinoma; NS, not significantly different.

## Data Availability

The datasets generated and/or analyzed during the current study are not publicly available but are available from the corresponding author upon reasonable request.

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
