# Peer review of "Insight into the Wnt Pathway in Sporadic Small Bowel Adenocarcinoma"

_cancers, 2025, doi:10.3390/cancers17182965_

Round 1

Reviewer 1 Report

Comments and Suggestions for Authors

The Introduction is somewhat lengthy and contains tentative statements (e.g., the quantities included in the APC) that could have been more appropriately reserved for the Discussion section.

To avoid repetition in the Introduction, you should not revisit the procedure, molecular subtypes, and previous literature reviews.

A clearer hypothesis or research question is needed at the end. A direct statement like "Our hypothesis is that..." would be a clear enhancement.

Immunohistochemistry: The criteria for "abnormal" and "decreased" expression could be better standardized across markers. Some thresholds (e.g., greater than 10% for nuclear β-catenin) appear arbitrary; were these based on previous literature?

The selection of 48 of 75 samples for sequencing should be explained; was this based on sample quality, availability, or selection bias? Sample selection should be clearly documented in the methodology section.

Multivariate model selection criteria should be briefly justified. In particular, it should be clearly documented why different models were generated in Table 3. Figure 1 (IHC images) is informative, but it would be helpful if each panel of images were labeled directly (A–H).

Statistical results: When reporting p-values, effect sizes or confidence intervals for correlations (e.g., the relationship between marker expression and clinical features) may provide better insight into clinical significance.

The discussion lacks a critical assessment of sampling bias: most tumors were duodenal; how might this affect generalizability?

It would be helpful to elaborate on the clinical implications of the findings: could Wnt pathway activation guide treatment or prognosis?

Some paragraphs too directly repeat the results (e.g., the Wnt5a findings); this space could be used to further interpret or critique the findings. The Discussion section should be edited to avoid repetition.

The potential dual role of Wnt5a (oncogenic and tumor suppressive) is noted but needs to be examined more critically, especially in light of contrasting findings in other cancers.

Conclusion: A brief discussion of future directions would strengthen the conclusion. Specifically, “Future studies should investigate whether targeting Wnt5a or restoring E-cadherin function could provide therapeutic benefit.”

A few citations in the introduction/discussion refer to multiple articles (e.g., "[3, 6-10]")—if some of these are review articles, it would be helpful to highlight this.

Ensure consistency in formatting (journal names, abbreviations, DOI availability).

Author Response

The Introduction is somewhat lengthy and contains tentative statements (e.g., the quantities included in the APC) that could have been more appropriately reserved for the Discussion section.

We agree with the reviewer’s comment and have moved the following sentence to the Discussion section. Page 9, lines 264-267: The relatively low incidence of APC mutations (7.1–37%) in sporadic SBA, despite their high frequency in adenomas, suggests that the adenoma-carcinoma sequence may not be a dominant pathway in SBA carcinogenesis.

To avoid repetition in the Introduction, you should not revisit the procedure, molecular subtypes, and previous literature reviews.

We agree with the reviewer’s comment. To avoid repetition in the introduction, we have removed the sections summarizing procedures, molecular subtypes, and previous literature reviews.

A clearer hypothesis or research question is needed at the end. A direct statement like "Our hypothesis is that..." would be a clear enhancement.

We added the following sentences in accordance with the reviewer’s comment. Page 11, lines 352-355: We propose that activation of the Wnt pathway has potential value in predicting treatment response and prognosis, as reduced expression of E-cadherin and β-catenin is associated with poorer outcomes and Wnt5a may represent a therapeutic target.

Immunohistochemistry: The criteria for "abnormal" and "decreased" expression could be better standardized across markers. Some thresholds (e.g., greater than 10% for nuclear β-catenin) appear arbitrary; were these based on previous literature?

We deleted the sentence “In addition, β-catenin nuclear expression was considered as aberrant, when ≥10% of cancer cells exhibited nuclear positivity.” and integrated β-catenin into the evaluation of cyclin D1 and c-Myc. The criterion for nuclear β-catenin positivity (≥10%) was based on previous literature (Wong SCC; Clin Cancer Res 2004). Page 4, lines 123-125.

The selection of 48 of 75 samples for sequencing should be explained; was this based on sample quality, availability, or selection bias? Sample selection should be clearly documented in the methodology section.

The selection was based on tissue availability. We added the following sentence. Page 4, lines 137-138: NGS was performed on 48 patients with sufficient DNA extracted from paraffin-embedded tissue blocks.

Multivariate model selection criteria should be briefly justified. In particular, it should be clearly documented why different models were generated in Table 3. Figure 1 (IHC images) is informative, but it would be helpful if each panel of images were labeled directly (A–H).

Yes, we agree with the reviewer’s comment. We combine the three models into a single model to simplify the results of the multivariate analysis (new Table 3). We labeled each panel directly in new Figure 2.

Statistical results: When reporting p-values, effect sizes or confidence intervals for correlations (e.g., the relationship between marker expression and clinical features) may provide better insight into clinical significance.

We examined correlations using the chi-square test or Fisher's exact test, which only allow us to determine the presence or absence of an association. Therefore, we cannot draw further conclusions regarding clinical significance.

The discussion lacks a critical assessment of sampling bias: most tumors were duodenal; how might this affect generalizability?

We added the following sentences to the Discussion section. Page 10, lines 341-345: Half of the cases in this study were located in duodenum. As the duodenum is the most common site of small bowel adenocarcinoma, this distribution is expected. However, the relatively high proportion of stage I cases in the duodenum may represent a potential source of bias. Nevertheless, because no significant differences in Wnt pathway molecules were observed by subsite, this is unlikely to affect the generalizability of our findings.

It would be helpful to elaborate on the clinical implications of the findings: could Wnt pathway activation guide treatment or prognosis?

Yes, we think it could. We added the following sentences in accordance with the reviewer’s comment. Page 11, lines 352-355: We propose that activation of the Wnt pathway has potential value in predicting treatment response and prognosis, as reduced expression of E-cadherin and β-catenin is associated with poorer outcomes and Wnt5a may represent a therapeutic target.

Some paragraphs too directly repeat the results (e.g., the Wnt5a findings); this space could be used to further interpret or critique the findings. The Discussion section should be edited to avoid repetition.

We deleted several paragraphs describing Wnt5a findings to avoid repetition.

The potential dual role of Wnt5a (oncogenic and tumor suppressive) is noted but needs to be examined more critically, especially in light of contrasting findings in other cancers.

We added the following sentences to the Discussion section. Page 10, lines 307-316: In colorectal cancer, Wnt5a has been reported to act as either an oncogene or a tumor suppressor, depending on downstream pathway activation [28]. Wnt5a can modulate both canonical and noncanonical Wnt signaling [29, 30], and its prognostic value appears to be context-dependent [28]. For example, Wnt5a has been shown to decrease Lgr5/RSPO3 expression and β-catenin activity in vitro, and its expression has been associated with improved survival in colon cancer [31]. In contrast, other studies have reported that Wnt5a promotes epithelial-mesenchymal transition and contributes to colorectal cancer progression and metastasis [32]. Similarly, Wnt5a gene expression has been associated with better overall survival in lung squamous cell carcinoma, but with poor prognosis in gastric adenocarcinoma [33].

Conclusion: A brief discussion of future directions would strengthen the conclusion. Specifically, “Future studies should investigate whether targeting Wnt5a or restoring E-cadherin function could provide therapeutic benefit.”

We added the following sentences to the Conclusions. Page 11, 355-356: Future studies should investigate whether targeting Wnt5a or restoring E-cadherin function could provide therapeutic benefit.

A few citations in the introduction/discussion refer to multiple articles (e.g., "[3, 6-10]")—if some of these are review articles, it would be helpful to highlight this.

We have referred to review papers as much as possible. In the case mentioned by the reviewer, no appropriate review paper was available, so we cited multiple original articles instead. We also deleted two older references.

Ensure consistency in formatting (journal names, abbreviations, DOI availability).

We edited the formatting in accordance with the reviewer’s comment.

Reviewer 2 Report

Comments and Suggestions for Authors

This study seeks to elucidate the function of Wnt pathway genes and proteins in small bowel adenocarcinoma by examining genetic alterations and protein expression in tumor tissues. Using immunohistochemistry and targeted next-generation sequencing, the researchers investigated Wnt pathway gene mutations (namely APC and CTNNB1) and the expression of downstream proteins (such β-catenin, cyclin D1, c-Myc, E-cadherin, and Wnt5a). They discovered that Wnt pathway protein activation (as seen by aberrant β-catenin, cyclin D1, c-Myc, and Wnt5a expression) was far more prevalent, occurring in up to 60% of cases and mutations in important Wnt pathway genes were only found in 23% of cases overall.  Not only the gene mutations highly correlate with aggressive illness and poor prognosis, but the study also finds that both canonical (β-catenin-dependent) and noncanonical (Wnt5a-related) Wnt signaling contribute to tumor growth and progression in SBA, especially the intestinal-type. Several areas require improvement to enhance clarity, scientific rigor, and readability of the manuscript. Major comments:

  1. I strongly encourage the authors to include a graphical abstract that reflects the study design results and conclusion.
  2. The authors should include time frame of sample collection in the materials and methods section.
  3. The authors should clearly mention how the 75 samples were distributed among all the mentioned adenocarcinomas, such as duodenal, jejunal, and ileal.
  4. What was the rationale for avoiding ampullary adenocarcinoma in the study?
  5. Did the authors assess the overall survival also, rather than just cancer-free survival?
  6. NGS was conducted in 48 samples out of 75. What is the reason behind this selection?
  7. Did the authors use any databases other than “ClinVar” for variant classification?
  8. Other than, β-catenin, cyclin D1, c-Myc, E-cadherin, and Wnt5a, consider expanding analyses to additional Wnt/β-catenin destruction complex genes (AXIN1/2, GSK3β, CK1) relevant to SBA pathogenesis.
  9. I suggest that the authors include a figure and a flowchart representing the study design. Such a figure would enhance the reader’s understanding by clearly illustrating the pathways and processes influenced, highlighting the role of Wnt pathway in small bowel adenocarcinoma.
  10. I suggest that the authors provide graphical survival displays (e.g., Kaplan–Meier curves with numbers at risk) since the current table is difficult to interpret.
  11. The authors should reorder the IHC narrative to match the sequence of images and panels provided.
  12. Justify the cutoffs for protein positivity (e.g., 10% nuclear staining) by referencing established standards or prior literature.

Minor comments:

  1. In section 2.4, the authors mentioned a database “CinVar” which should be corrected as “ClinVar”. Such mistakes should be thoroughly proofread by the authors to aid the reproducibility and clarity of the study. 
  2. Throughout the manuscript, full forms should be mentioned when discussing that particular word for the first time, thereafter, relevant abbreviations should be provided. 
  3. The authors should provide a separate section for all abbreviations used in the study. 
  4. In table-1 and table-2, “NS” abbreviation should be given as not significantly different for better understanding. 
  5. Similarly in table-3, abbreviations should be given clearly under the table for clarity.

Author Response

Major comments:

  1. I strongly encourage the authors to include a graphical abstract that reflects the study design results and conclusion.

In accordance with the reviewer’s comment, we added two graphical abstracts: Figure 1 (study flowchart) and Figure 5 (graphical summary of the results and conclusions).

  1. The authors should include time frame of sample collection in the materials and methods section.

We added the time frame of sample collection in the Materials and Methods section as follows. Page 3, line 92: We analyzed 75 tissue samples of duodenal, jejunal and ileal adenocarcinoma obtained from the archives of the Department of Pathology at Nippon Medical School Hospital and the Department of Pathology at Nippon Medical School Chiba Hokusoh Hospital between January 2006 and December 2022.

  1. The authors should clearly mention how the 75 samples were distributed among all the mentioned adenocarcinomas, such as duodenal, jejunal, and ileal.

We added the following sentences in accordance with the reviewer’s comment. Page 4, lines 160-163: Tumors were located in the duodenum in 40 cases, the jejunum in 30 cases and the ileum in 5 cases. Mucin immunophenotypes included gastric (n=14), gastrointestinal (n=30), intestinal (n=29) and null-type (n=2).

  1. What was the rationale for avoiding ampullary adenocarcinoma in the study?

The ampulla of Vater is the structure in the duodenal wall where the biliary and pancreatic ducts open. It is lined by pancreatobiliary-type mucosa, while its outer surface is lined by enteric-type mucosa. Owing to this complex anatomy, tumors arising in the ampulla represent a heterogeneous group. For this reason, we distinguished ampullary adenocarcinoma from small bowel adenocarcinoma. 

  1. Did the authors assess the overall survival also, rather than just cancer-free survival?

All patients died of small bowel adenocarcinoma; therefore, the overall survival and cancer-specific survival outcomes were identical.

  1. NGS was conducted in 48 samples out of 75. What is the reason behind this selection?

This selection was based on tissue availability. We added the following sentences. Page 4, lines 138-139: NGS was performed on 48 patients with sufficient DNA extracted from paraffin-embedded tissue blocks.

  1. Did the authors use any databases other than “ClinVar” for variant classification?

No, we didn’t.

  1. Other than, β-catenin, cyclin D1, c-Myc, E-cadherin, and Wnt5a, consider expanding analyses to additional Wnt/β-catenin destruction complex genes (AXIN1/2, GSK3β, CK1) relevant to SBA pathogenesis.

We performed NGS using the Ion AmpliSeq Cancer Hotspot Panel v2, and did not detect any gene mutations other than APC and CTNNB1 among Wnt pathway -related molecules. Therefore, we believe there are no abnormalities in other Wnt/β-catenin destruction complex genes, including AXIN1/2, GSK3β, CK1.

  1. I suggest that the authors include a figure and a flowchart representing the study design. Such a figure would enhance the reader’s understanding by clearly illustrating the pathways and processes influenced, highlighting the role of Wnt pathway in small bowel adenocarcinoma.

In accordance with the reviewer’s comment, we added two graphical abstracts Figure 1 (study flowchart) and Figure 5 (graphical summary of the results and conclusions). In addition, we included a graphical illustration of the correlation between the involvement of each molecule in the Wnt pathway, as inferred from our results.

  1. I suggest that the authors provide graphical survival displays (e.g., Kaplan–Meier curves with numbers at risk) since the current table is difficult to interpret.

Yes, we agree with the reviewer’s comment. We combined the three models into a single model to simplify the results of the multivariate analysis and to facilitate easier interpretation by readers (Table 3).

  1. The authors should reorder the IHC narrative to match the sequence of images and panels provided.

In accordance with the reviewer’s comment, we reordered the IHC narrative to match the sequence of images and panels. Page 7, lines 202-207.

  1. Justify the cutoffs for protein positivity (e.g., 10% nuclear staining) by referencing established standards or prior literature.

We integrated β-catenin into the evaluation of cyclin D1 and c-Myc. The criterion for nuclear β-catenin positivity (≥10%) was based on previous literature (Wong SCC; Clin Cancer Res 2004).

Minor comments:

  1. In section 2.4, the authors mentioned a database “CinVar” which should be corrected as “ClinVar”. Such mistakes should be thoroughly proofread by the authors to aid the reproducibility and clarity of the study. 

We apologize the typo. We have thoroughly proofread the manuscript to ensure that no such errors remain and to enhance the reproducibility and clarity of the research.

  1. Throughout the manuscript, full forms should be mentioned when discussing that particular word for the first time, thereafter, relevant abbreviations should be provided. 

In accordance with the reviewer’s comment, we revised the relevant abbreviations.

  1. The authors should provide a separate section for all abbreviations used in the study. 

In accordance with the reviewer’s comment, we provided a separate section for all abbreviations in the text, Figures and Tables. Page 11, lines 376-377.

  1. In table-1 and table-2, “NS” abbreviation should be given as not significantly different for better understanding.
  2. Similarly in table-3, abbreviations should be given clearly under the table for clarity.

In accordance with the reviewer’s comment, we added not significantly as abbreviation of “NS” in Table 1 and Table 2, and added abbreviations under Table 3. 

Reviewer 3 Report

Comments and Suggestions for Authors

The authors have designed and presented the work in a clear way and it is of high significance. I have few comments which the authors needs to address:

  1. the title of the article looks more like for review and thus I suggest to modify the title the manuscript.
  2. Figure 1 could have been more informative if the annotations could have been used in the image and I know that most of the image analysis softwares have this option.
  3. The names of the proteins are in some places with capital letter and in some places in small letters, please make it consistent and in case of human it is mostly written with capital letters.
  4. In Table 1, please use * or make it bold for those where p-value is significant.
  5. For figure 2, could you please crosscheck with TCGA database.

Author Response

1. the title of the article looks more like for review and thus I suggest to modify the title the manuscript.

We agree with the reviewer’s comment. We changed the title of the manuscript to “Insight into the Wnt pathway in sporadic small bowel adenocarcinoma.”

2. Figure 1 could have been more informative if the annotations could have been used in the image and I know that most of the image analysis softwares have this option.

Since we did not use image analysis software, we were unable to apply that option. However, following Reviewer 1's suggestion, we labeled each image with the corresponding protein name.

3. The names of the proteins are in some places with capital letter and in some places in small letters, please make it consistent and in case of human it is mostly written with capital letters.

If the protein name is written in capital letters, it may be confused with the gene name, so we kept the current format. Our writing style is consistent with that of most papers; for example, articles published in Cancers also use ‘cyclin D1’. (Cancers 2023 Oct 18;15(20):5032. doi: 10.3390/cancers15205032).

4. In Table 1, please use * or make it bold for those where p-value is significant.

We agree with the reviewer’s comment and have highlighted values with significant p-values in bold.

5. For figure 2, could you please crosscheck with TCGA database.

We attempted to cross-check using TCGA database; however, due to the rarity of small bowel adenocarcinoma, sufficient data were not available.

Round 2

Reviewer 2 Report

Comments and Suggestions for Authors

This study elucidated the role of Wnt pathway genes and proteins in small bowel adenocarcinoma by analyzing genetic alterations and protein expression in tumor tissues. The authors have satisfactorily addressed all reviewer comments during the revision process, and the manuscript is now suitable for acceptance and publication.